# Reliable repurposing of the antibody interactome inside the cell

Caitlin M. O'Shea[1], Rushba Shahzad[1], Kimia Aghasoleimani[1], Stuart Newman[1], Jiraporn Panmanee[2], Leonard C. Schalkwyk[1], Greg N. Brooke[1], Fiona E. Benson[3], James S. Trimmer[4], Daryl A. Bosco[5], Takao Fujisawa[6,7], Hidenori Ichijo[6,7], Neil R. Cashman[8], Stanislav Engel[9] & Gareth S. A. Wright[1] ✉

Eighty-five percent of the human proteome has at least one interacting monoclonal antibody. These molecules penetrate the cytoplasm poorly and are very often non-functional within the cell. Analysis of antibody variable domains and characterisation of forty-five single-chain variable fragment (scFv) intrabodies expressed in human cells indicated charge to have the greatest impact on solubility. We created new interdomain linkers, optimised scFv domain orientation and found an optimisable charge discrepancy between variable heavy framework and CDR sites. When applied to reduce the search space and rank the products of AI-led inverse folding this creates a single highly soluble, abundant and stable intrabody with parent antibody epitope recognition. Over six hundred intrabody sequences are presented targeting sixty cytoplasmic proteins with linear, conformational, post-translational modification or oligomer specificity. Interactions were validated for p53, α-synuclein, SOD1, polyQ, FUS/TLS, UCHL1 and GFP. Here we show reliable repurposing of the sequenced antibody interactome inside the cell.

Antibodies are the precision arm of the immune system, as well as valuable research tools and medicines. More than $10^9$ antibodies have been sequenced[1] and over $10^5$ are defined with a known antigen[2]. Antibodies are composed of heavy and light chains that are translated into the endoplasmic reticulum. After assembly they are transported through the Golgi, during which they are glycosylated and form intra- and inter-domain disulphide bonds, before secretion. Extracellular native and therapeutic antibodies can be internalised through endocytic pathways but are degraded in lysosomal compartments with few accounts of effective direction to the cytoplasm[3,4]. This limits our ability to deploy the antibody interactome inside the cell.

Pairs of antibody variable domains housing complementarity determining regions (CDRs) can be peptide linked to form single-chain variable fragments (scFvs)[5,6] (Fig. 1a, b) and expressed within cells as intrabodies. However, scFv intrabodies also have notoriously low solubility leading to aggregation and loss of function[7]. The development of new synthetic biology applications could be propelled by the ability to generate custom intracellular biological interactors. Grafting CDR loops onto variable domain framework regions with desired

[1]School of Life Sciences, University of Essex, Colchester, United Kingdom. [2]Research Center for Neuroscience, Institute of Molecular Biosciences, Mahidol University, Nakhon Pathom, Thailand. [3]Division of Biomedical and Life Sciences, Faculty of Health and Medicine, Lancaster University, Lancaster, United Kingdom. [4]UC Davis/NIH NeuroMab Facility, Department of Physiology and Membrane Biology, University of California School of Medicine, Davis, CA, USA. [5]UMass Chan Medical School, Worcester, MA, USA. [6]Laboratory of Cell Signalling, Graduate School of Pharmaceutical Sciences, The University of Tokyo, Bunkyo-ku, Tokyo, Japan. [7]Cell Signalling and Stress Responses Laboratory, Advanced Research Initiative, Institute of Integrated Research, Institute of Science Tokyo, Tokyo, Japan. [8]Djavad Mowafaghian Centre for Brain Health, University of British Columbia, Vancouver, BC, Canada. [9]Department of Clinical Biochemistry and Pharmacology, Faculty of Health Sciences, Ben-Gurion University of the Negev, Beer-Sheva, Israel. ✉e-mail: gareth.wright@essex.ac.uk

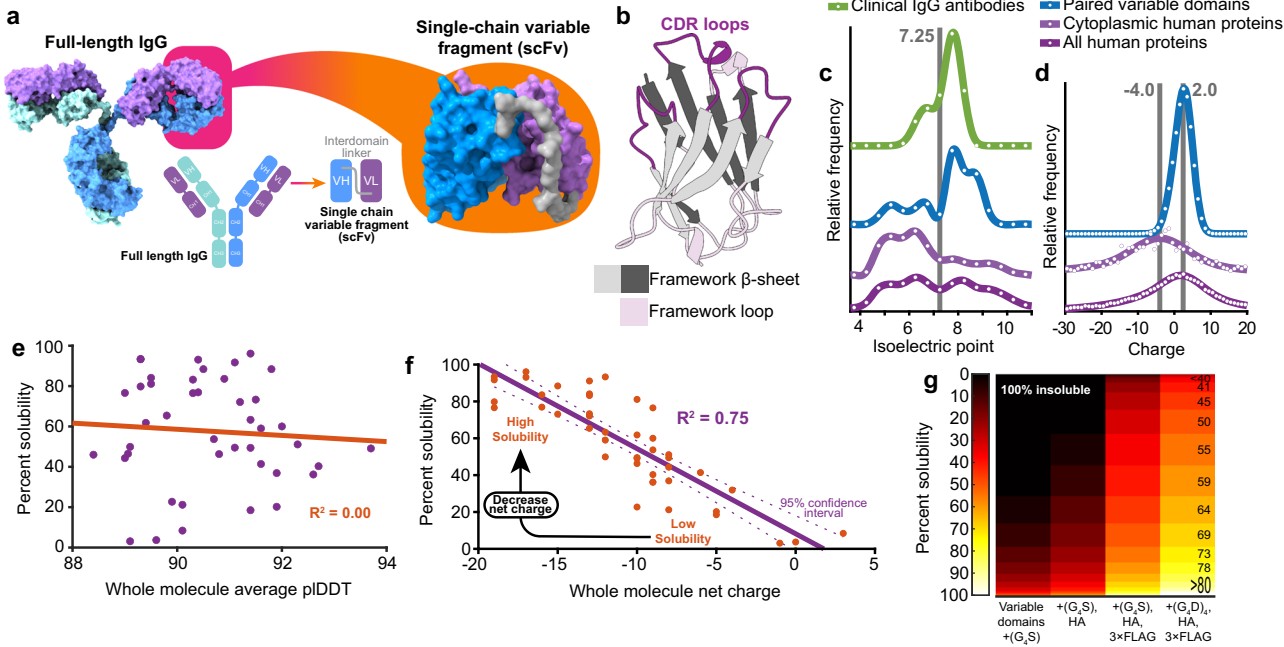

**Fig. 1 | Whole molecule net charge is a determinant of scFv intrabody solubility.** **a** Structural and schematic representations of antibody to scFv reformatting with variable domains (VL and VH) joined by a peptide linker (grey). Complete IgG image PDB 1IGT. **b** Cartoon representation of an antibody variable domain showing CDR and framework regions. **c** Isoelectric point of full-length clinical IgG antibodies, paired variable domains, human cytoplasmic proteins and the complete human proteome. A characteristic minimum at physiological pH is highlighted in grey. Extracellular proteins predominate in the 7.5–11 region while intracellular proteins are found 4–7.5. **d** Paired variable domains have a different charge profile in comparison with cytoplasmic proteins. Mode 2.0 and −4.0 (highlighted grey), mean 1.8 ± 2.6 and −8.7 ± 28.3 respectively, mean ± one standard deviation. **e** Average pLDDT scores calculated from AlphaFold3 structural models have no correlation with scFv intrabody solubility. **f** scFv solubility correlates with net charge. Data points are mean solubility values for 45 scFv intrabodies, $P = 4.4 \times 10^{-14}$ (two-tailed f-test) with 95% confidence level indicated. Source data are provided as a Source Data file. **g** Intracellular solubility prediction for 68,551 scFv with various modifications to the interdomain linker, N and C-termini.

characteristics[8], engineering physicochemical properties by rational mutagenesis[9], removing cysteines that form disulphide bonds and directed evolution[10] have been applied to this problem but no strategy consistently facilitates antibody to intrabody reformatting. Maintenance of specificity and affinity from parent antibody to derived scFv intrabody is important but paratope–epitope interactions can only operate if a correctly structured scFv can exist in and diffuse through the cell. High solubility, thermal stability and demonstrable epitope binding are therefore the key characteristics of deployable scFv intrabodies and should be the focus of a broadly applicable reformatting strategy.

Here we define the intracellular solubility of forty-five scFvs that have applications in the study and treatment of neurodegenerative diseases including Alzheimer's, Parkinson's, Huntington's diseases and amyotrophic lateral sclerosis. We use these data to discover the determinants of scFv intrabody solubility and develop a system to predict scFv solubility based on molecular charge. In addition, we describe new interdomain linkers that contribute to intracellular solubility. From this work emerges a rationale for scFv intrabody design that ensures first construct success making the use of these molecules reliable and economical. Applying our approach to the products of AI-led molecular redesign allowed in silico reformatting of 672 non-redundant antibodies as scFv intrabodies targeting intracellular proteins including tau, α-synuclein, SOD1, TDP-43, p53, HIF-1α, histones and ubiquitin along with protein phosphorylation, citrullination and acetylation post-translational modifications. The sequences of all intrabodies described are made available following the principles of open source science[11]. This should aid design and implementation of synthetic molecular pathways based on collocation of component macromolecules and therapeutics able to interact with protein sub-populations known to co-segregate with disease all within the living cell.

## Results

### Variable domain charge drives intrabody solubility

The immunoglobin (Ig) protein fold superfamily is widely found throughout nature. It predates the existence of vertebrate antibodies and its members are often found at high concentrations within cells[12–14]. For example, Cu,Zn-superoxide dismutase has an Ig fold, is near ubiquitous throughout life, abundant and very often found in the cytoplasm where it maintains a disulphide bond[14,15]. Conversely, scFv expression in the cytoplasm often results in poor solubility and functionality. To find why variable fragments (Fv) are poorly adapted to the intracellular environment we compared the physicochemical properties of the complete and cytoplasmic human proteomes with $10^6$ paired variable light (VL) and heavy (VH) domains from mice, rats and humans. Variable domains exist within the bounds of hydrophobicity and stability common in the cytoplasm (Supplementary Fig. 1a–d). Figure 1c shows extracellular proteins, including full-length antibodies and variable domains, are likely to have an isoelectric point (pI) above 7.4. Seventy percent of Fvs are found in this region and are ill adapted to a role as an scFv intrabody. Charge strongly influences pI and only 0.4% of the human proteome with a charge of zero or less has a pI over 7.4 (Supplementary Fig. 1e). Direct comparison of net charge indicates antibody variable domains are generally more electropositive than cytoplasmic proteins (Fig. 1d).

We defined the solubility of forty-five scFv intrabodies and compared that experimental data to predictions made by nine sequence-based solubility calculators (Supplementary Fig. 2). The ProteinSol algorithm predicted scFv solubility well ($R^2$ 0.715) using a weighted combination of protein features focusing on charge and hydrophobicity[16]. We calculated 79 individual physicochemical characteristics for each protein aiming to unpick the drivers of scFv intrabody solubility and find a single, easily modifiable

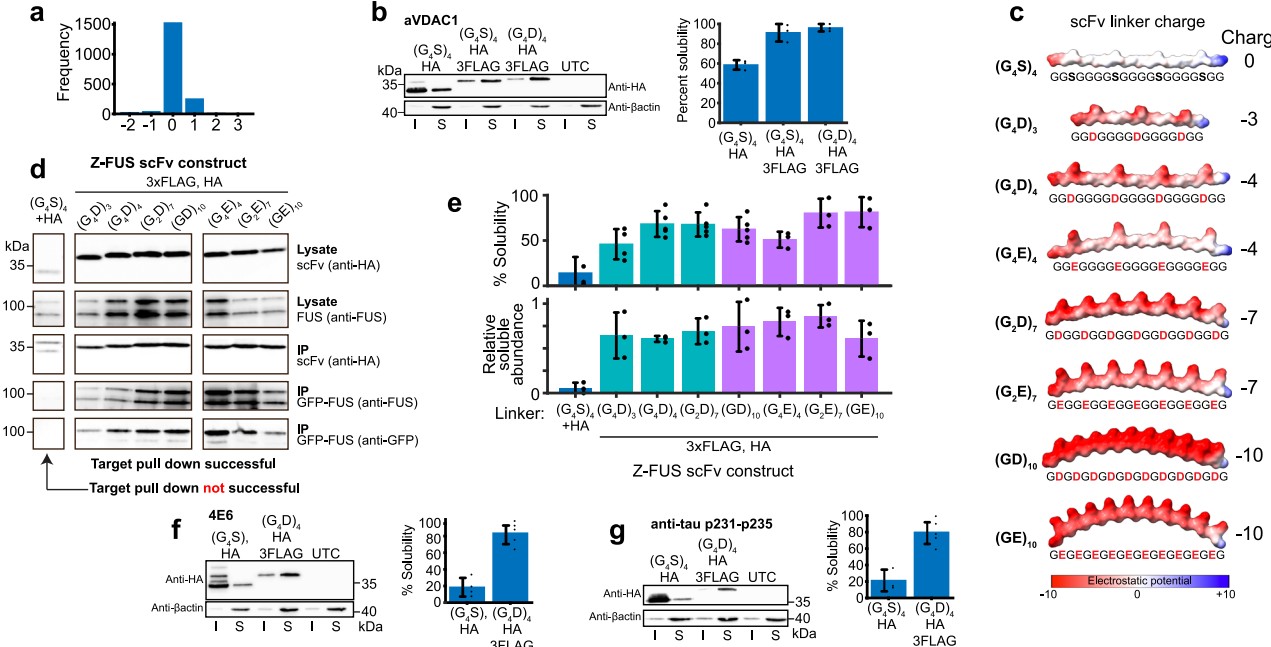

**Fig. 2 | Charged interdomain linkers improve scFv intrabody solubility.**
**a** Frequency bar chart showing the charge of 1840 scFv interdomain linker sequences in the NCBI database. **b** Increased solubility of an anti-VDAC1 scFv with charged interdomain linker (upper). Sample processing controls (lower), $n = 3$ biological replicates, $(G_4S)_4$/HA vs $(G_4S)_4$/HA/3FLAG $P = 0.001$, $(G_4S)_4$/HA/3FLAG vs $(G_4D)_4$/HA/3FLAG $P = 0.43$ (two-tailed, two-sample equal variance $t$ test).
**c** Coulombic electrostatic potential colouring of negatively charged interdomain linker structural representations. **d** Co-immunoprecipitation of GFP-FUS/TLS protein by Z-FUS-derived scFv intrabodies with varying charge inter-domain linkers bound by immobilised anti-HA beads and visualised by SDS-PAGE/western blot. Antibodies used to probe western membranes are stated to the right. Images showing anti-GFP and anti-FUS blots were run on separate gels. **e** Percent solubility

and abundance in the soluble fraction of scFv intrabodies shown in (**d**), $n = 3$ biological replicates, $(G_4S)_4$/HA vs $(G_4D)_4$/HA/3FLAG solubility $P = 0.0007$, abundance $P = 0.0002$ (two-tailed, two-sample equal variance $t$ test). Blue – serine linker, green – aspartate linkers, purple – glutamate linkers. **f**, **g** High solubility engineered into scFv intrabodies, derived from 4E6 ($P = 0.0001$, two-tailed, two-sample equal variance $t$ test) and anti-p231-235 ($P = 0.0016$, two-tailed, two-sample equal variance $t$ test) antibodies, targeting tau with combined N- and C-terminal tags and linker modifications, 4E6 $n = 4$ and 5 biological replicates. p231–235 $n = 3$ and 5 biological replicates. Lower panels are sample processing controls. All error bars represent one standard deviation. Source data are provided as a Source Data file. I – Insoluble fraction, S – Soluble fraction.

characteristic (Supplementary Data 1). Hydrophobicity, aliphatic index, charge or isoelectric point of CDR loops, individual variable domains or their framework regions in addition to predicted thermal stability[17] and whole molecule predicted local difference distance test (pLDDT) score calculated from AlphaFold3 structural models were poor predictors of solubility (Fig. 1e, Supplementary Table 1). The latter was included because AlphaFold pLDDT score is a good predictor of disorder[18,19]; the antithesis of stability. Whole molecule net charge at physiological pH has a negative linear correlation with solubility over the range +3 to −20 ($R^2$ 0.75) (Fig. 1f), exceeding correlations with related characteristics including charge at pH 5.5[20] ($R^2$ 0.69), pI ($R^2$ 0.39), and all solubility calculators (Supplementary Fig. 2). This relationship is implemented in the bioinformatics website scFvright which isolates Fv protein sequences from full-length antibodies or takes full-length scFv with fusion tags and linkers then predicts the percentage likely to be found in the soluble fraction (https://scfvright.essex.ac.uk) (Supplementary Software 1).

**Large scale prediction of scFv intrabody solubility**
Using the relationship presented in Fig. 1f, we predicted the intracellular solubility of 68,551 Fv and scFv proteins. Only 0.02% of unmodified Fvs would have high solubility within the cell, which we have defined as over 70% partitioning in the soluble mammalian cell fraction (Fig. 1g). Addition of canonical Gly-Gly-Gly-Gly-Ser ($G_4S$) linkers and a C-terminal HA tag yields 0.03% with high solubility reflecting the historical problem of scFv expression in the cytoplasm. Kabayama et al. introduced the combination of 3xFLAG and HA tags to improve scFv

solubility and monodispersity in the cell[20] by reducing net charge by −9, but still only 3.9% are highly soluble.

**Electronegative linkers improve scFv intrabody solubility**
High scFv solubility requires a net charge of less than −15 (Fig. 1f). However, the pre-eminence of distributed negative charge means Asp/Glu residues can be incorporated or Arg/Lys residues removed in any location that does not affect target binding or structural stability; ideally in solvent exposed linker sites, fusion tags or framework sites outside the paratope. Ninety-seven percent of scFvs contain interdomain linkers that do not contribute to, or increase, net charge (Fig. 2a). Ninety-two percent are $G_4S$-type linkers[6] or closely related sequences that are not optimised for expression of intracellular fusion proteins. Modification of the $G_4S$ sequence to include aspartate or glutamate in place of serine to yield $(G_4D)_4$ or $(G_4E)_4$ and contribute to solubility by adding four negative charges (Fig. 2b, c). Linkers $(G_2D)_7$, $(G_2E)_7$, $(GD)_{10}$ and $(GE)_{10}$ also add negative charge (Fig. 2c). Each provides sufficient flexibility to maintain target binding to FUS/TLS protein by an scFv intrabody derived from Z-FUS-5 antibody (Fig. 2d). Assessment of scFv linker variants indicate each confers different properties with $(G_2E)_7$ having the highest segregation into, and abundance in, the soluble fraction (Fig. 2e). Therefore, different combinations of serine, aspartate and glutamate in the linker would allow tuning of scFv negative charge from zero to −10 in addition to native Fv and fusion tag charge. These linkers require little expertise to implement and in combination with 3xFlag and HA tags are predicted to confer greater than 70% and 50% solubility on 22% and 84% of scFv

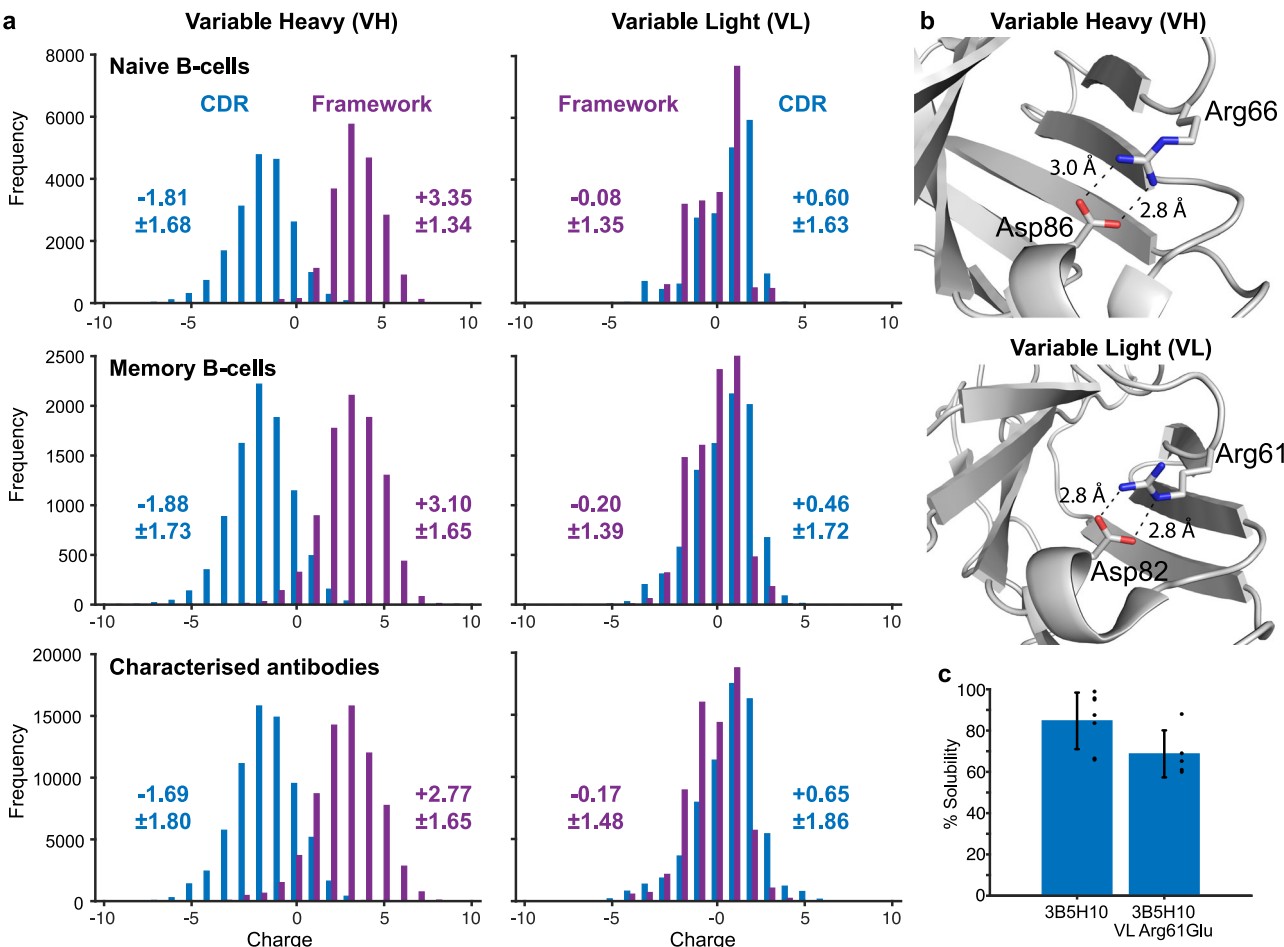

**Fig. 3 | Antibody variable domain CDR loop and framework region charge. a** The majority of variable heavy domain CDR loops are negatively charged and normally distributed around −1.81 to −1.69 dependent on maturation state. VH framework regions are predominantly positively charged and distributed around +2.77 to +3.10. This effect is observed irrespective of variable domain numbering scheme (Supplementary Table 2). Conversely, light chain CDR and framework charge display overlapping distributions. A slight skew to positive and negative respectively is conserved between Chothia and Kabat but not IMGT numbering schemes (Supplementary Table 2). **b** Salt-bridge bonding between β-strands D and F of the 3B5H10 antibody variable domains. Bonding distances annotated. **c** Charge substitution mutation VL Arg61Glu reduces 3B5H10-derived scFv intrabody solubility by breaking the salt-bridge in **b**. Error bars represent one standard deviation, $n = 7$ and 5 biological replicates, $P = 0.058$ (two-tailed, two-sample equal variance $t$ test). Source data are provided as a Source Data file.

intrabodies respectively when using $(G_4D)_4$ or $(G_4E)_4$ versions (Fig. 1g). This is illustrated by the high solubility of scFv intrabodies with charged linkers derived from antibodies targeting tau (Fig. 2f, g).

## Modifications to framework sites improve solubility

Aspartate scanning through CDR loops has been shown to reduce variable domain aggregation[21] but is unfavourable given the possibility that CDR mutations may change epitope specificity or affinity. To find an alternative approach we analysed the charge of framework and CDR (Fig. 1b) sequences from 166,179 Fv molecules (Fig. 3a). VH domain CDR and framework sequences have characteristic, non-overlapping charge frequency distributions. VH CDR loops are generally electronegative while VH framework regions are electropositive. VH dipolarity marginally reduces as antibodies mature but is absent in VL domains (Fig. 3a). Arginine and lysine residues are not found within the core Ig-fold β-sandwich but very often project into solvent, except for a ubiquitous salt-bridge between VL domain Arg/Lys61-Asp82 and VH domain Arg/Lys66-Asp86 (Chothia numbering) found at the N-terminus of β-strands D and F (Fig. 3b). Mutations at this site in an scFv intrabody derived from the 3B5H10 poly-glutamine specific antibody[22] reduce solubility likely by reducing thermal stability (Fig. 3c). With this exception, positively charged amino acids in variable domain framework sites, particularly VH, may be good sites to introduce negative charge.

## scFv domain order effects intrabody solubility

scFvs are typically designed with VH domain N-terminal to the VL domain (VHVL)[6]. However, variable heavy domains are known to have heterogenous physical and chemical properties, including thermal stability[23], in comparison with VL domains. Analysis of fifteen scFv intrabodies in both VHVL and VLVH orientations shows they have similar solubility when the net charge of the VH domain is zero or less (74.1 ± 17.3% and 72.8 ± 17.2% respectively, mean ± one standard deviation). However, VHVL scFv intrabodies with positively charged VH domains have reduced solubility with respect to the VLVH form (49.3 ± 21.9% and 58.2 ± 3.5% respectively, mean ± one standard deviation) (Supplementary Fig. 3). Variable heavy domain charge is negatively correlated with overall VHVL scFv solubility ($R^2$ 0.38) but poorly corelates with VLVH solubility ($R^2$ 0.15) (Supplementary Fig 3b). A positively charged VH domain often seems to require molecular chaperoning by its accompanying VL domain which cannot happen if it is translated first. VLVH scFv intrabodies have overall better solubility and are more reliable therefore, but this effect can be overcome by charge swap mutations that increase net charge of the VH domain (Supplementary Fig. 3c).

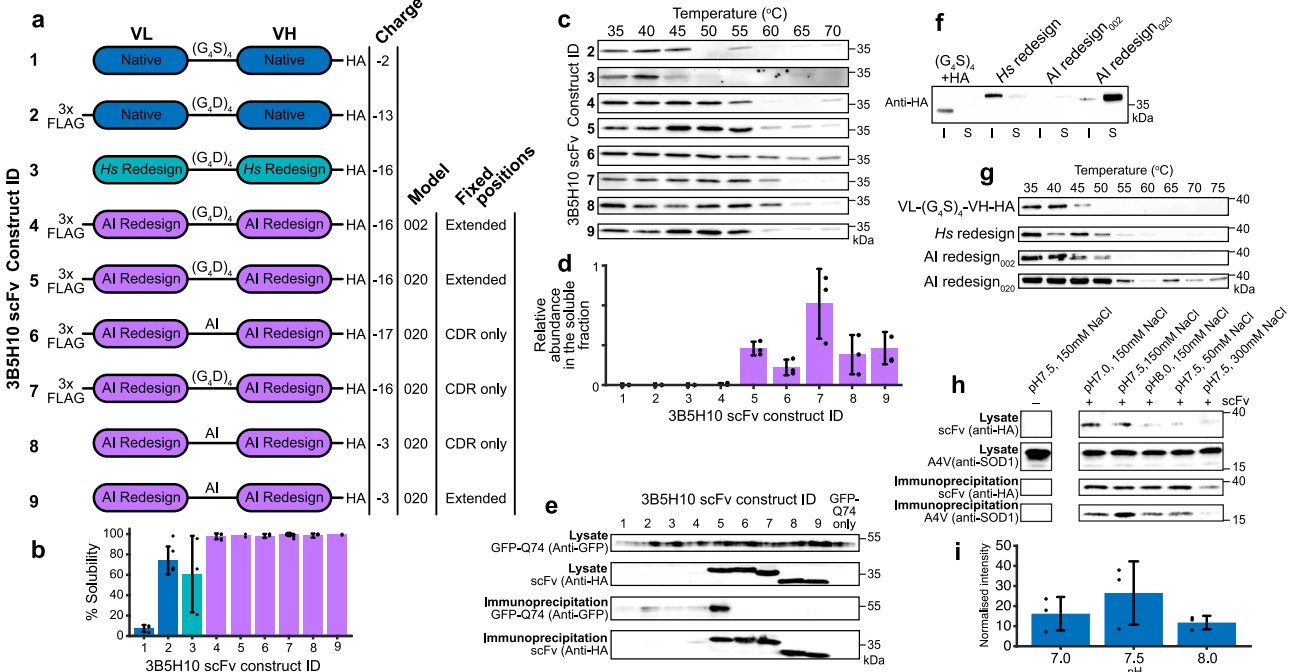

**Fig. 4 | Inverse folding improves scFv intrabody solubility and thermal stability. a** 3B5H10-derived scFv intrabodies created using native, human redesigned (VL K20D, VL K42D, VH K13D, VH K19D, VH K23E), and ProteinMPNN$_{SOL}$ inverse folded variable domains with stated tags, linkers and model noise. Fixed residue positions CDR only: VL 24–34, 50–56, 89–97 and VH 25–32, 52–56, 95–102. Extended: VL 24–36, 47–56, 86–97 and VH 26–59, 91–105. **b** Percent of 3B5H10-derived scFv found in the soluble cell fraction. $n = 3, 6, 3, 5, 2, 5, 5, 6, 2$ biological repeats. **c** In-cell thermal stability assay of 3B5H10-derived scFv intrabodies. Stability for construct 1 could not be measured. Representative of $n = 2$ biological replicates. **d** Abundance of 3B5H10-derived scFv in the soluble fraction. $n = 3$ biological replicates. **e** Co-immunoprecipitation of GFP-polyQ74 by 3B5H10-derived scFv intrabodies bound by immobilised anti-HA beads and visualised by SDS-PAGE/

western blot. Antibodies used to probe western membranes are stated to the left. **f** SDS-PAGE/western blot showing the solubility and **g** Intracellular thermal stability of MS785-derived scFv intrabodies. **h** Co-immunoprecipitation of A4V SOD1 by MS785-derived scFv (AI redesign$_{020}$) intrabody bound by immobilised anti-HA beads and visualised by SDS-PAGE/western blot. **i** Quantification of A4V SOD1 immunoprecipitation by MS785 showing robust binding across pH 7.0–8.0 with a non-significant preference for physiological pH, $n = 3$ biological replicates, $P = 0.284$ (one way ANOVA with Dunnett's post hoc analysis). Antibodies used to probe western membranes are stated to the left. Chothia numbering throughout. Error bars represent one standard deviation. $Hs$ – human, AI – ProteinMPNN$_{SOL}$ inverse folding. Source data are provided as a Source Data file. Western blots are representative of at least two experiments.

## scFv intrabody reconstruction by inverse folding

Data point residuals in Fig. 1f indicate charge is not the only factor that influences scFv intrabody solubility. We used ProteinMPNN$_{SOL}$[24] to increase scFv intrabody thermal stability and solubility while implementing domain orientation and charge rules to restrict design options and filter outputs respectively for poly-glutamine-specific antibody 3B5H10-derived[22] scFv intrabodies (Fig. 4a). The inverse folding approach creates protein sequences that are able to satisfy the geometric constraints of a given protein backbone structural model[25,26] and ProteinMPNN$_{SOL}$ is known to increase surface hydrophilicity[24]. All inverse folded scFv intrabodies display near complete solubility and high thermal stability in comparison with native and human redesigned constructs (Fig. 4b, c). However, sequences produced with model structure coordinate noise of 0.20 Å in comparison with 0.02 Å have higher abundance in the soluble fraction (Fig. 4d). Increased abundance and thermal stability has been noted previously on inverse folded proteins using 0.20 Å model noise expressed in bacteria[27]. Increased model noise is inversely proportional to sequence recovery[28] which may facilitate beneficial changes. Minimally restricting fixed residue positions to only CDR loops or allowing redesign of the interdomain linker did not produce protein capable of interaction with co-expressed GFP-polyQ (Fig. 4e, constructs 6-9). Variable domain interfaces affect antigen binding[29]. This may be as part of an extended epitope-paratope interaction surface[30] or through domain, and therefore CDR loop orientation, given two thirds of the β-sheets N- and C-terminal to each CDR loop are found in the interface. As CDR loops very often lack structure until interaction

with an epitope we chose to focus on maintenance of domain orientation. Fixing CDR loops, linker and domain interface β-sheets again produced highly soluble, abundant and stable proteins but only one strongly immunoprecipitated GFP-polyQ (Fig. 4e, construct 5). Sequences of each construct can be found in Supplementary Data 2.

To test reproducibility, we reformatted the misfolding-specific anti-SOD1 antibody MS785[31] as a scFv intrabody. A traditional $(G_4S)_4$ + HA tag construct, or a human redesigned variant with charge swap mutations, were both insoluble (Fig. 4f). Inverse folding with 0.02 Å or 0.20 Å model noise produced highly soluble proteins with the latter abundant in the soluble fraction; >4000-fold abundance increase and ~10 °C increase in melting temperature in comparison with the best human design (Fig. 4f, g). This MS785 scFv variant reliably pulls down A4V SOD1 with an optimum at physiological pH and NaCl concentration (Fig. 4h, i).

## Impact of inverse folding on scFv intrabodies

Effective scFv construction and redesign using the rules established above (Fig. 5a) can be applied to any antibody. Accordingly, scFv intrabodies were produced by inverse folding from 672 Fv sequences with specificity for cytoplasmic human proteins and oligomeric or post-translationally modified states thereof (Fig. 5b). Across 672,000 derivative scFv sequences no position was consistently substituted in all inverse folding outputs but some are predisposed to changes, including domain N- and C-termini (Fig. 5c) and surface sites, for example, small hydrophobic amino acids totalling 98.1% of original sequences at highly solvent exposed VH position 11 are substituted for

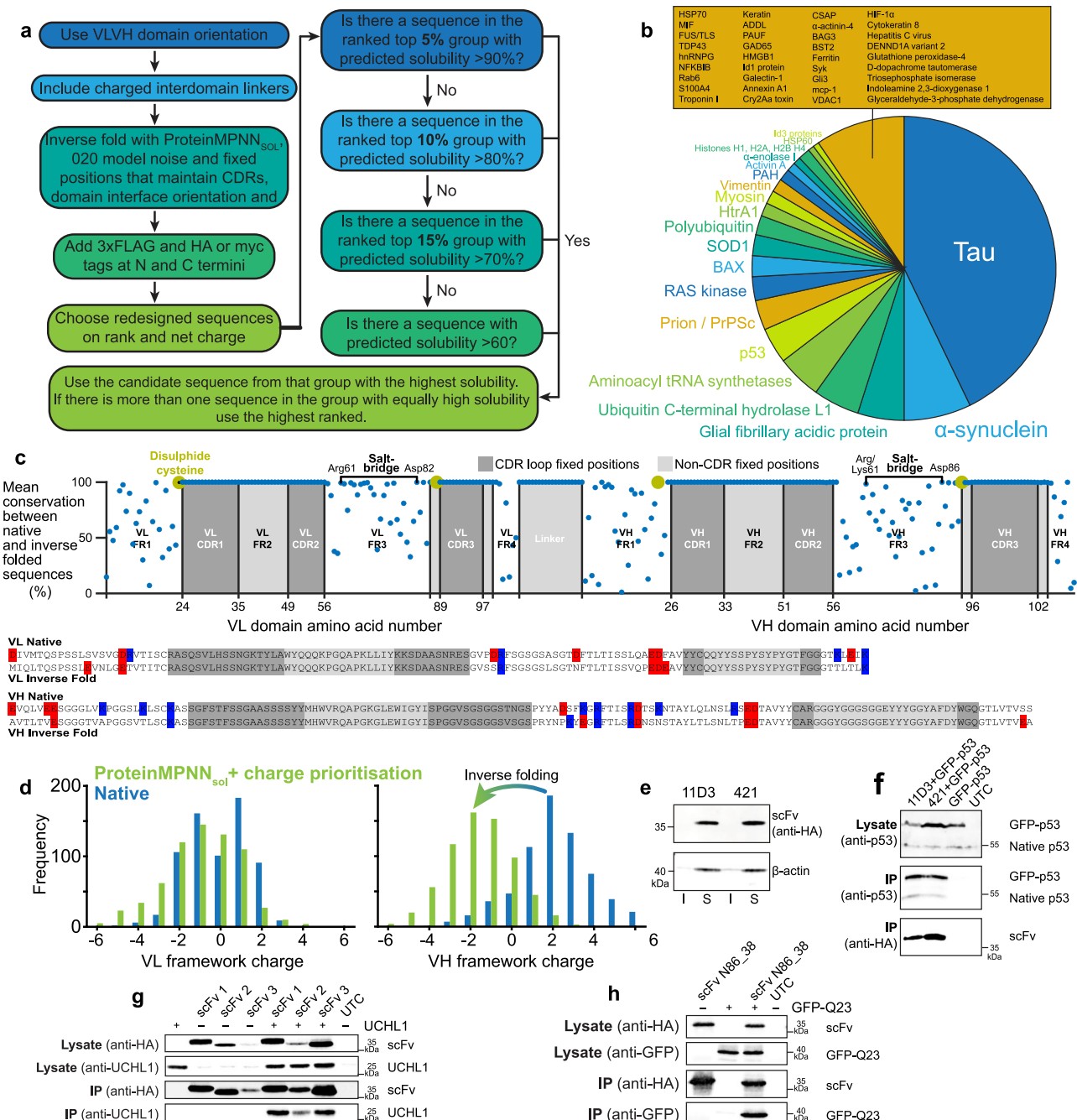

**Fig. 5 | Reliable in silico construction of scFv intrabodies. a** Design rules for reformatting an antibody as an optimised scFv intrabody. **b** Intracellular target specificity of 672 scFv constructed using the principles in (**a**). **c** Mean amino acid conservation for native to inverse folded scFv intrabody and charged residues in non-fixed positions within variable domain consensus sequences. **d** Charge frequency histograms for VL and VH framework sites before and after inverse folding. **e** Anti-p53 scFv 11D3 and 421 constructed using rules in (**a**) are highly soluble. Lower pane are sample processing controls. **f** Co-immunoprecipitation of p53 by anti-p53 scFv intrabodies. **g** Co-immunoprecipitation of UCHL1 by anti-UCHL1 scFv intrabodies. **h** Co-immunoprecipitation of GFP by anti-GFP scFv intrabody derived from Neuromabs antibody N86_38. All immunoprecipitation visualised by SDS-PAGE/ western following immobilisation with anti-HA beads. Antibodies used to probe western membranes are stated to the left. IP Immunoprecipitation. Chothia numbering throughout. Western blots are representative of at least two experiments.

polar or charged amino acids in 93.9% of inverse folded variants. Conversely, disulphide bonding cystines and salt-bridge charged residues are strictly maintained with also VL amino acids 61–69 and VH 149, 151–154 showing strong conservation through inverse folding. The consensus sequences of parent and inverse folded progeny show a preference for substitutions in VH domains using 0.02 Å model (16 % and 32% of non-fixed VL and VH positions respectively) but this discrepancy is reduced when using the 0.20 Å model (31% and 34% respectively). Furthermore, changes incorporated by model 0.20 Å

reduce mean net charge of VH framework sites from $+2.1 \pm 1.8$ to $-1.6 \pm 1.6$ (mean ± one standard deviation). This reduces the number of VH domains with net positive charge framework regions from 83.9% to 9.3% (Fig. 5d). This effect is not seen for VL domain framework sites that are natively better adapted to the cytoplasmic environment (Fig. 5d and 3a). As a result of inverse folding with ProteinMPNN$_{sol}$ using 0.20 Å model, consensus VH domain net charge is reduced from +1 to −1 and VL maintained at +2. Conversely, model 0.02 Å increases VL domain net charge from +2 to +5 and maintains VH charge. This

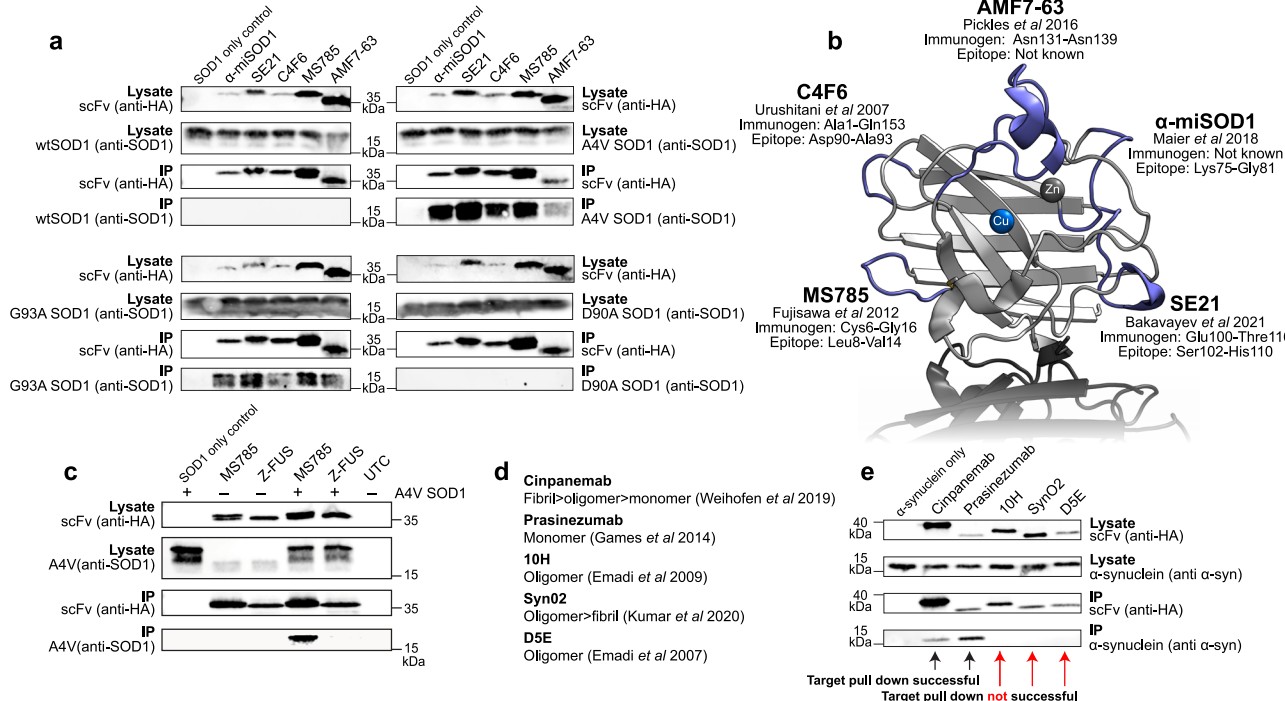

**Fig. 6 | State specific intrabody interactions with SOD1 and α-synuclein.**
**a** Assessment of the interaction between scFv intrabodies derived from anti-SOD1 conformation specific antibodies with wild-type, A4V, G93A and D90A SOD1 by co-immunoprecipitation on anti-HA beads and visualised by SDS-PAGE/western blot. Antibodies used to probe western membranes are stated on the left and right.
**b** SOD1 conformation specific intrabodies target five distinct epitopes in the SOD1

monomer here shown in the fully metalated disulphide intact form.
**c** Immunoprecipitation of A4V SOD1 by MS785-derived scFv but not Z-FUS-derived scFv. **d** Oligomeric state specificity of five anti-α-synuclein antibodies. **e** Co-immunoprecipitation of α-synuclein by scFv intrabodies derived from antibodies in (**d**) on anti-HA beads and visualised by SDS-PAGE/western blot. Antibodies used to probe western membranes are stated. IP Immunoprecipitation.

effect observed using 0.20 Å model is mediated by a reduction of the total number of charged residues but disproportionately reducing the number of positively charged amino acids (Fig. 5c) and is mediated by changes to only forty-four (15 %) of the total amino acid positions. Of those forty-four, forty substitutions (90 %) are to amino acids with side chains largely or entirely solvent exposed sites. Two of the remaining four substitutions are found on a loop (VL Met4Leu) or the end of a β-strand away from the core β-sandwich (VL Leu78Val), and two maximise hydrophobic packing around the β-sandwich core (VH Glu6Val and VH Ile69Leu). These may contribute to increased VH domain thermal stability.

Two intrabodies derived from antibodies 11D3 and 421 targeting the C-terminus of p53 tumour suppressor protein were tested to validate the redesign process. Unmodified scFv derived from 11D3 and 421 antibodies are predicted to have 40% and 30% solubility respectively based on Fig. 1f. Previous work showed scFv 421 to express well but with staining resembling aggregation whereas no expression was observed for scFv 11D3[32]. By contrast, redesign yields good expression with very high solubility (>99%) (Fig. 5e) and both intrabodies maintain parent antibody interactions with p53 (Fig. 5f). As further validation, we created three new highly soluble intrabodies for ubiquitin carboxy-terminal hydrolase 1 (UCHL1) (Fig. 5g) and one for GFP (Fig. 5h) and validated their target interactions. Sequences are reported in Supplementary Data 4.

### State specific recognition
Repurposing well characterised conformation and post-translational state specific antibodies that interact with structures that co-segregate with neurodegenerative disease is an advantage of our approach. Figure 6a shows interaction assessments for five conformational specific scFv intrabodies derived from parent antibodies shown to be specific for forms of SOD1 found in familial or sporadic ALS. C4F6 scFv

intrabody has reactivity to ALS-associated G93A and A4V mutants but not wild-type or wild-type-like D90A replicating the specificity of the parent antibody[33,34]. Intrabodies derived from AMF7-63, α-miSOD, SE21 and MS785 antibodies follow the same pattern[31,35–37]. This is despite each intrabody interacting with a distinct and separate loop epitope within the SOD1 molecule (Fig. 6b). Inverse folded intrabody specificity was determined using the non-SOD1 specific Z-FUS intrabody with (G4E)4 linker which was found not to pull down A4V SOD1 (Fig. 6c). Sequences are reported in Supplementary Data 5.

Antibodies with a proclivity for binding to the different quaternary structures presented by α-synuclein[38–42] are reactive to epitopes found in Parkinson's disease and other α-synucleinopathy tissues but have also been the subject of many clinical trials[43,44] (Fig. 6d). Reformatting as intrabodies (Supplementary Data 6) and comparison of α-synuclein interaction by co-immunoprecipitation following co-expression indicates the monomeric form prevails in our HEK293T expression system with oligomer or fibril states either absent or below the limit of detection (Fig. 6e). This work highlights the ability of these interactors to differentiate an intracellular protein sub-population and will surely expand to recognition of post-translational state and heterocomplex species.

## Discussion
The relationship between protein charge and solubility is well characterised[45] and introducing negatively charged mutations or fusion tags has been shown to improve scFv behaviour[20,46,47]. We take this concept to its conclusion by showing scFv intrabody solubility has a negative, linear relationship with net charge (Fig. 1f) and to achieve ≥80% solubility an scFv intrabody should have charge less than −15. Calculated integer net charge is likely a proxy for surface charge. The size of antibody variable domains, their folding around a hydrophobic core and association across a hydrophobic interface mean all variable

domain charged residues are at least partially solvent accessible, but solubility predictions may be improved by greater granularity in surface charge calculations.

It is very unlikely a charged protein will have an isoelectric point equivalent to physiological pH (Supplementary Fig. S1e). That equivalence is avoided in the human proteome (Fig. 1c, d) due to the likelihood of self-association and aggregation[48]; any charge is better than no charge. However, positively charged proteins interact with nucleic acids and cell membranes thereby inhibiting free diffusion and monodispersity. DNA and RNA are negatively charged and it may be that most cytoplasmic proteins have evolved a similar charge profile to prevent unwanted interactions. In the extracellular space that selective pressure is absent but is replaced by the necessity to diffuse through the charged mesh of proteoglycans. This manifests as very different isoelectric point and charge distribution profiles between intracellular and variable domain protein sequences (Fig. 1c, d). Nevertheless, charge is not the only contributor to protein solubility. High thermal stability prevents exposure of hydrophobic, aggregation-prone residues normally sequestered in the molecular core. This is well articulated by inverse folded 3B5H10-derived scFv variants with high thermal stability, solubility and near neutral charge, but these molecules do not retain epitope interactions (Fig. 4, constructs 8 & 9). We found that introducing positive to negative charge swap mutations had a cumulative effect on solubility but compounding five or six mutations reduced protein abundance. This is indicative of reduced thermal stability and resulting propensity for degradation. By contrast, inverse folding with ProteinMPNN$_{SOL}$ changed ~30% of non-fixed positions, increased thermal stability and solubility, while maintaining function. Evidently it is possible to break out of strictures imposed by the inverse relationship between stability and function.

The scFv intrabodies with experimentally defined solubility described here (Supplementary Data 1) were created from monoclonal antibody sequences found in public databases or retrieved by hybridoma DNA or mass spectrometry protein sequencing. They comprise a range of molecules with native net charge −6 to +6 representative of 95% of the charge distribution presented in Fig. 1d. The documented antibody interactome can now be effectively repurposed inside the cell. While that interactome is large, it is the result of in vivo generation; antigen presentation in animals, hybridoma technology, B-cell sequencing and two-hybrid screening that are limited by scale. De novo antibody design, specifically prediction of CDR-epitope interactions is being revolutionised[49–51], but currently requires production and testing of a prohibitively large numbers of putative interactors. This process will certainly become more efficient and accessible. The scFv design rules established here ensure first construct design success and provide an access route for either in vivo or in silico generated antibodies targeting any molecule within the cell. That may be interactions with conformationally plastic or intrinsically disordered proteins that are difficult to address with small molecules, protection of functional sites, inhibition of protein-protein interactions, molecular chaperoning, enzyme inhibition, providing substrate recognition for biological protein editors like recently described for tau and SOD1[52,53], or the molecular interactions to collocate elements of new enzymatic pathways. The intrabodies described are derived from antibodies used in research but also those that have been tested in clinical trials that did not meet end point goals. Neurodegenerative protein misfolding disease immunotherapies in particular are plagued by this limited efficacy. Bringing those interactions inside the cell where disease related changes occur may expedite medicine development especially given the burgeoning effectiveness, acceptance and safety of gene therapies. This approach also capitalises on the large amount of research that has already been performed on those molecules in a manner analogous to small molecule drug repurposing. Introduction of synthetic molecules into humans and other animals will require careful consideration and

testing of their immunogenicity, however we note a recent report of a ProteinMPNN variant that has been tuned to reduce MHC-I visibility[54].

## Methods

### Expression plasmid construction mutagenesis
All intrabody coding sequences were synthesised by Twist Bioscience and cloned into Twist Bioscience's CMV BetaGlobin WPRE Neo plasmid vector without coding for sequences for N-terminal localisation signals but including, where stated, N- and C-terminal tags and interdomain linkers. Site directed mutagenesis was performed using Platinum SuperFi II DNA Polymerase and sequence verified.

### Solubility assay
HEK293T cells (ATCC CRL-3216) were cultured in DMEM medium with 10% foetal calf serum supplemented with 100 units/ml penicillin and 100 µg/ml streptomycin in six-well plates. Cells were transfected with intrabody expression plasmids using Turbofect reagent and harvested after 24 h. Media was removed, 300 µl phosphate buffered saline (PBS) with protease inhibitor cocktail (Roche EDTA-free, cOmplete) was added and cells lifted with gentle pipetting. Cells were lysed by sonication at 4 °C. Soluble and insoluble cell fractions separated by centrifugation at 20,000 × g for 45 min at 4 °C. The insoluble component was resuspended in 300 µl PBS. Reducing and denaturing SDS-PAGE was performed and proteins transferred to nitrocellulose membrane by dry western blotting. Membranes were probed with anti-HA (clone 6I21 Proteintech 81290-1, clone 1F5C6 Proteintech 66006-2), anti-FLAG (clone 8H6A10 Proteintech HRP-66008) or anti-myc (clone 4A6 Merck 05-724), anti-FUS/TLS (clone 1B4F8 Proteintech 68262-1), anti-SOD1 (clone 2F10G1 Proteintech 67480-1), anti-α-synuclein (clone 1B10E9 Proteintech 66412-1), anti-UCHL1 (clone 1C9E11 Proteintech 66230-1) antibodies with appropriate secondary antibodies (LI-COR IRDye 680RD goat anti-rabbit 926-68071, IRDye 800CW goat anti-mouse 926-32210, Proteintech HRP-conjugated goat anti-mouse SA00001-1 or Invitrogen HRP-conjugated goat anti-rabbit A16104). All antibodies were used at 1:2000 dilution in 50 mM Tris-HCl pH 7.4, 150 mM NaCl, 0.05% Tween-20.

### Immunoprecipitation
Immunoprecipitation was performed at room temperature. HEK293T cells were cultured and transfected as above. Cells were lifted in 25 mM Tris-HCl pH 7.4, 150 mM NaCl, 1% NP-40, 1 mM EDTA, 5% glycerol with protease inhibitor, incubated for 30 min and centrifuged at 12,000 × g for 10 min. Supernatant was applied to pre-equilibrated HA-Trap magnetic agarose beads (Proteintech ATMA-200) and washed with PBS with protease inhibitor cocktail then eluted by heating to 95 °C for 5 min in SDS-PAGE sample buffer. SDS-PAGE and western blot was performed on the samples. PageRuler prestained protein ladder (Thermo #26616) was used throughout.

### Cellular thermal shift assay
HEK293T cells were cultured and transfected as above then resuspended in 400 µl PBS with protease inhibitor cocktail and 40 µl aliquoted into PCR tubes. Cell suspensions were heated at 35, 40, 45, 50, 55, 60, 65, 70, 75 °C for 5 min then frozen at −80 °C. Cells were then thawed and sonicated and centrifuged at 20,000 × g for 45 min at 4 °C and the supernatant heated with SDS-PAGE loading buffer before SDS-PAGE and western blot.

### Hybridoma DNA and mass spectrometry protein sequencing
Mouse anti-TDP43 C-terminus DB9 antibody and rat anti-SOD1 MS785 antibody expressing hybridomas were sequenced at The Pirbright Institute, UK. Briefly, 5´-Rapid amplification of cDNA ends-ready cDNA was prepared followed by PCR amplification of the heavy and light chain Ig variable regions. The products were subsequently cloned into a sequencing vector and Sanger sequenced. Variable domain

sequences can be found in Genbank PV987755.1, PV987756.1, PV987757.1 and PV987758.1. C4F6 SOD1 conformational specific antibody was previously sequenced by tryptic digest mass spectrometry[55].

## In silico protein physicochemical analysis

In silico physicochemical analysis was performed on 1,073,713 paired variable domain sequences retrieved from the Observed Antibody Space (OAS) database[56]; 843 proteins annotated as cytoplasmic only localisation by the Human Protein Atlas[57] and 20,590 proteins representing the complete, but non-isoformic, human proteome (UP000005640) downloaded from Uniprot[58]. The isoelectric point of 127 full-length clinically approved IgG antibodies of multiple subtypes was also calculated. Isoelectric point and peptide mass were calculated with standalone Isoelectric Point Calculator[59] taking the average value for pI. Charge at physiological pH was calculated as the difference between the number of arginine and lysine residues and aspartate and glutamate; $(K+R)-(D+E)$. Charge at pH 5.5 was calculated with the addition of the number of histidines; $(K+R+H)-(D+E)$. Grand average of hydropathy (GRAVY) was calculated as the ratio of the count of each amino acid scaled using Kyte & Doolittle paramaterisation[60] to the total number of amino acids. Aliphatic index was calculated as the mole percent fraction of alanine, valine, isoleucine and leucine amino acids[61]. Mean pLDDT was calculated as the mean of all atom pLDDT scores from variable domains and linkers. DeltaG IF and DeltaG (Kcal/Mol) were predicted with using the method of Cagiada et al.[17].

## scFvright website

scFvright predicts scFv solubility based on the formula: percent solubility $= -4.6237 \times$ net charge $+ 8.2469$, derived from Fig. 1f. The operation can be performed on Fv protein sequences isolated from larger Fab or full-length antibody sequences, which uses AbRSA to isolate variable domain sequences, or scFv protein sequences including linkers and fusion tags. The website is available at https://scfvright.essex.ac.uk.

## CDR and framework charge analysis

CDR and framework sequences were isolated using AbRSA[62] from 68,551 non-identical variable domain sequences taken from the Patent & Literature Antibody Database (PLAbDab)[2], 8993 antibody-coding memory B cells[63] and 19303 naïve B cells[64]. Chothia numbering[65] was used to determine CDR-framework boundaries throughout using AbRSA[62], but IMGT and Kabat procedures were also applied to memory B cell data to ensure observed charge distribution differences were not an artefact of a particular numbering scheme. Mean CDR and framework charge for all three numbering schemes are presented in Supplementary Table 2.

## scFv interdomain linker charge analysis

The NCBI databases were searched using the term 'scFv' yielding 9688 sequences. Those with a pair of variable domains and a linker length longer than six amino acids, 1840 sequences, were accepted for analysis. Linker sequences were extracted with AbRSA[62] and checked to ensure no framework residues were incorporated at N and C termini. Charge at neutral pH was calculated as above.

## Human redesign of scFv to maximise net charge

Sequences of AMF763, 3B5H10 and MS785 derived scFv targeting polyQ amd SOD1 were used to predict structural models with AlphaFold3. They were redesigned by incorporating positive to negative charge swap mutations at sites where lysine and arginine residues were clearly solvent exposed and deemed to not make side chain interactions with any other part of the molecule. 3B5H10 scFv mutations VL K20D, VL K42D, VH K13D, VH K19D, VH K23E and MS785 scFv mutations VL K18D, VL K45D, VH K15D (Chothia numbering).

## High throughput in silico scFv intrabody construction

The PLAbDab was filtered to remove antibodies targeting non protein targets, bacterial, viral, extracellular, membrane proteins, bi- or tri-specific antibodies, those with no defined target, unusually long CDR loops or where AbRSA[62] failed to assign three CDRs and four framework regions per variable domain. Remaining sequences were then sifted for CDR loop sequence redundancy with the lowest charge paired variable domain chosen over other isoforms. This yielded 672 sequences. Net charge was calculated as above along with predicted solubility of scFv including triple FLAG tag, HA tag and $(G_4D)_4$ linkers. scFv protein structures were predicted with AlphaFold3[66] on an Nvidia A100 40GB GPU machine running CUDA 12.6. Inverse folding was performed with ProteinMPNN$_{SOL}$[24,26] using 0.20 Å and 0.02 Å models directed to design 100 sequences while fixing amino acids in VL and VH domains corresponding to CDR1, framework region 2, CDR2 and CDR3 extended by three residues at both N and C termini defined using AbRSA[62] and corresponding to Chothia numbering VL24-56, VL86-100, linker, VH26-56, VH92-105. Inverse folded outputs were then sorted based on predicted solubility calculated using the formula % solubility $= -4.6237 \times$ net charge $+ 8.2469$ in combination with ProteinMPNN rank (described in Fig. 5a).

## Analysis of ProteinMPNN$_{SOL}$ outputs

To generalise the effect of inverse folding using ProteinMPNN$_{SOL}$ on this large amount of structurally similar proteins all sequences were aligned, substitution mutations were analysed and compared between outputs using 0.20 Å and 0.02 Å models. Six hundred and seventy-seven variable domains including linkers were aligned with Clustal Omega with Pearson/FASTA output to create a 294-character alignment. One hundred ProteinMPNN$_{sol}$ inverse folded output sequences for each of the 672 parent scFv were then padded using the aligned parent as a template. This allowed quantification of the number of positions conserved between ProteinMPNN$_{sol}$ output sequences in comparison with each parent scFv. Consensus sequences for 673 parent scFv and 672,000 inverse folded sequences, using 020 and 002 models, were calculated with Jalview.

## Reporting summary

Further information on research design is available in the Nature Portfolio Reporting Summary linked to this article.

## Data availability

All scFv intrabody sequences described can be found in Supplementary Data 1–6. Source data are provided with this paper including raw western blot images and quantifications as source data files. Source data are provided with this paper.

## Code availability

scFvright website code is made available as a separate file (Supplementary Software 1) or upon request to authors.

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

## Acknowledgements

We thank Prof. Yang Cao (Center of Growth, Metabolism and Aging, Key Laboratory of Bio-Resource and Eco-Environment of Ministry of Education, College of Life Sciences, Sichuan University, Chengdu, 610065, People's Republic of China) for the use of AbRSA. Funding: National Research Council of Thailand (NRCT) (N42A670122) J.P.; National Institutes of Health (U24 NS109113, U24 NS119916) J.S.T.; JSPS (25H00971), JST (JPMJMS2022-18), AMED (JP21gm5010001, 22gm0010009s0101, 23gm0010009s0102, 23gm1710001s0102) H.I.; Israel Science Foundation (grant no. 221/22) S.E.; Royal Society (RGS\R1\231005), Academy of Medical Sciences (SBF008\1028), Motor Neurone Disease Association (Wright/Oct18/969-799, Wright/Apr25/2472-791) G.S.A.W.

## Author contributions

C.M.O.'S. performed formal analysis, investigation, visualisation and original draft preparation and editing. R.S. performed investigation and contributed to manuscript editing. K.A. contributed to investigation and manuscript editing. S.N. performed code construction and manuscript editing. J.P. provided resources and contributed to manuscript editing. L.C.S. provided supervision and manuscript editing. G.N.B., provided supervision and manuscript editing. F.E.B., provided resources and contributed to manuscript editing. J.S.T., provided resources, contributed to manuscript editing and funding acquisition, D.A.B., provided resources. T.F., Provided resources. H.I., Provided resources and funding acquisition. N.R.C., provided resources, funding acquisition and manuscript editing. S.E., provided resources, funding acquisition and manuscript editing. G.S.A.W., performed conceptualisation, formal analysis, funding acquisition, investigation, visualisation, methodology, project administration, code construction, supervision, and original draft preparation and editing.

## Competing interests

N.R.C. is Chief Science Officer of ProMIS Neurosciences, and a Professor Emeritus at the University of British Columbia. The remaining authors declare no competing interests.
