## [Transparent Peer Review file · Nature Communications]

Reliable repurposing of the antibody interactome inside the cell

Corresponding Author: Dr Gareth Wright

Version 0:

Reviewer comments:

Reviewer #1

(Remarks to the Author)

This manuscript presents a systematic approach to converting antibodies into functional intrabodies through charge optimization and computational redesign. The work addresses a significant technical challenge in the field and provides both methodological advances and a substantial resource of 672 redesigned intrabody sequences. The authors effectively identify the core issue - that antibody variable domains have charge distributions unsuited for the cytoplasmic environment - and provide a quantitative solution. The linear relationship between net charge and solubility ($R^2 = 0.75$, Fig. 1f) offers a simple, predictive framework. The testing of 45 scFv intrabodies with varied specificities provides robust experimental support. The inclusion of disease-relevant targets (SOD1, α -synuclein, p53, tau) enhances translational relevance. There are several novel contributions, including introduction of charged interdomain linkers (G₂D₇, G₂E₇, etc.) is elegant and practical, domain orientation insights (VL-VH vs VH-VL) provide actionable design guidelines, and the integration with ProteinMPNN_sol demonstrates effective use of AI tools. Additionally, making 672 intrabody sequences publicly available represents a valuable community resource, particularly for neurodegenerative disease research. This work makes important contributions to intrabody engineering and provides practical solutions. The design rules are immediately applicable, and the resource of pre-designed intrabodies will accelerate research. However, the impact would be strengthened by addressing the following areas:

1. Functional Validation Gap: While solubility is extensively characterized, functional binding is only demonstrated for a subset. The assumption that CDR preservation maintains binding affinity needs broader validation, and no quantitative affinity measurements (K_D values) are provided. Recommend including SPR or BLI data for key intrabodies comparing parent antibody vs. redesigned scFv affinities.
2. Statistical Analysis and Reproducibility: Many experiments show $n \geq 2$ or $n=3$ without power analysis justification. Error bars are inconsistently defined (standard deviation vs. SEM). Figure 4h lacks quantification despite showing a key functional validation. Statistical tests for significance are absent throughout. Please include statistical methods section, perform appropriate tests (t-tests, ANOVA), report exact p-values, and justify sample sizes.
3. Structural Analysis Limitations: Heavy reliance on AlphaFold3 predictions without experimental structure validation. The 0.02 Å vs 0.20 Å noise parameter choice lacks clear rationale. No discussion of potential structural perturbations from extensive mutagenesis. The work will be enhanced to include at least one crystal structure or solution NMR validation of a redesigned intrabody.
4. Immunogenicity: ProteinMPNN designed FW sequences deviate from natural human antibody V genes, discussion around immunogenicity mitigation should be included.
5. Charge Calculation Oversimplification The (K+R)-(D+E) calculation ignores histidine contributions and local pK_a shifts. Consider using more sophisticated electrostatic calculations.
6. Co-IP Controls Figure 6 co-immunoprecipitation experiments lack proper negative controls (empty vector, irrelevant scFv).

Minor Issues

- Supplementary files organization could be improved with a summary table
- Some references are incomplete (e.g., ref 47, 48 listed as preprints)
- The scFvright website functionality should be described in more detail

Reviewer #2

(Remarks to the Author)

This manuscript presents a comprehensive study on the physicochemical determinants of scFv intrabody solubility. The authors identify net charge as the dominant predictor of intracellular solubility and establish a linear model (scFvright) correlating charge and solubility ($R^2=0.75$). They further employ AI-based inverse folding using ProteinMPNNSOL to redesign scFv sequences with improved solubility and thermal stability. Overall, the study provides mechanistic insights and a potential design framework for intracellular antibodies.

The topic is timely and relevant, combining empirical biophysical characterization with AI-driven protein design. However, several methodological and interpretational aspects require clarification or further validation before publication.

Major Comments

1. Benchmarking of the scFvright model.

The scFvright tool was trained on 45 scFv samples and yielded an R^2 of 0.75 between solubility and net charge. To support the robustness of this relationship, the authors should benchmark their linear model against existing solubility predictors such as Protein-Sol [1], DeepSol [2], or Protsolm [3]. Comparative analysis would confirm whether the proposed linear regression is sufficiently accurate and whether charge alone can capture most solubility variation.

2. Feature interaction analysis.

Although the manuscript reports that hydrophobicity, pLDDT, and other physicochemical features individually correlate weakly with solubility, it would be valuable to assess possible interaction effects with charge. A multivariate regression or feature interaction model could reveal whether, for instance, charge-hydrophobicity coupling provides additional explanatory power.

3. Clarification of scFvright's purpose

The scFvright web tool appears to serve only as a validation of the empirical charge-solubility relationship. Since it is not subsequently used in the AI design workflow, its independent contribution to the study is unclear. The authors should clarify whether scFvright is intended solely for exploratory prediction or as part of the broader intrabody engineering pipeline.

4. Interpretability of ProteinMPNNSOL results

The inverse folding section demonstrates improved solubility and thermal stability, but lacks analysis explaining why certain AI-generated sequences perform better than others. Comparing top-ranked (e.g., top 1%) versus low-ranked (e.g., bottom 50%) sequences could identify key sequence or structural determinants—such as altered surface charge, salt bridges, or hydrophilic patches—that underlie the observed performance gap.

5. Reproducibility and workflow transparency

The AI redesign approach is innovative but under-documented. A complete workflow should be provided, detailing the parameters used (e.g., fixed residues, linker constraints, noise level, selection criteria, number of generated variants). Supplementary tables or pseudo-code would substantially improve reproducibility and future applicability of ProteinMPNNSOL for antibody optimization.

6. Integration of functional prediction tools

While fixing framework or linker regions helps preserve antigen binding, it also limits AI design freedom. The authors could enhance the design pipeline by integrating antibody binding or affinity prediction tools to prescreen ProteinMPNNSOL generated variants for binding competence. This would establish a more complete and automated workflow from sequence design to functionality prediction.

References

- [1] Hebditch M, Carballo-Amador M A, Charonis S, et al. Protein-Sol: a web tool for predicting protein solubility from sequence. *Bioinformatics*, 2017, 33(19): 3098–3100.
- [2] Khurana S, Rawi R, Kunji K, et al. DeepSol: a deep learning framework for sequence-based protein solubility prediction. *Bioinformatics*, 2018, 34(15): 2605–2613.
- [3] Tan Y, Zheng J, Hong L, et al. Protsolm: Protein solubility prediction with multi-modal features. *Proc. IEEE BIBM*, 2024: 223–232.

Version 1:

Reviewer comments:

Reviewer #1

(Remarks to the Author)

Thank you for making the edits and addressing the questions.

Reviewer #2

(Remarks to the Author)

Thank you for the comprehensive responses. All six of my points have been satisfactorily addressed; the additional

benchmark data, workflow details and clarifications on scFvright's scope fully answer my concerns. I have no further requests.

University of Essex

WHERE CHANGE HAPPENS

Wednesday 19th November 2025
RE: NCOMMS-25-55178

Please find below our responses to referees comments for our paper 'Reliable repurposing of the antibody interactome inside the cell' NCOMMS-25-55178. We thank the reviewers for their very positive feedback and we appreciate they recognised the value of our paper. Please find below our responses to each reviewer comment:

Reviewer #1

1. Functional Validation Gap: While solubility is extensively characterized, functional binding is only demonstrated for a subset. The assumption that CDR preservation maintains binding affinity needs broader validation.

We have included new immunoprecipitation data for three intrabodies that bind to UCLH1 (**Fig. 5g**) and one that interacts with GFP (**Fig. 5h**). Each was constructed using the approach described. There are now seventeen intrabodies with validated interactions against different epitopes across seven proteins including multiple epitopes within SOD1, α -synuclein and UCHL1 in the paper. As noted in the text we do not only preserve CDR loops; β -strands throughout the interface are also maintained. This means more than half of the residues are purposely fixed while the inverse folding process retains the original amino acid in more than 85 % of sites across the consensus scFv sequence. We have added a note in the text to make this clear.

No quantitative affinity measurements (KD values) are provided. Recommend including SPR or BLI data for key intrabodies comparing parent antibody vs. redesigned scFv affinities.

No quantitative affinity measurements are available in the literature for the majority of the parent antibodies used as templates for our intrabodies. Gathering that data for both parent antibody and derived intrabody would incur a very significant time and financial cost beyond the scope of this project.

2. Statistical Analysis and Reproducibility: Many experiments show $n \geq 2$ or $n = 3$ without power analysis justification. Error bars are inconsistently defined (standard deviation vs. SEM).

We purposely avoided SEM as it reflects variance of the mean rather than a measure of the data distribution. All error bars and error measurements throughout the paper represent one standard deviation. Those instances where this was not explicitly stated have been corrected. Sample size was based on established practice within the field. Straightforward, stable chemical measurements like this have small coefficients of dispersion and experience indicates that pair or triplicate repeats are adequate. Sample size numbers are reported in the appropriate figure legends and instances where these were omitted have been corrected.

Figure 4h lacks quantification despite showing a key functional validation. Statistical tests for significance are absent throughout. Please include statistical methods section, perform appropriate tests (t-tests, ANOVA), report exact p-values, and justify sample sizes.

We have included quantification of the abundance of SOD1 bound by MS785 across pH range 7.0 to 8.0 as **Figure 4i**. P values have been included where relevant with a description of statistical tests included in Method section. As is customary for molecular biology, experiments are repeated a minimum of two or three times. Repeat numbers are stated in figure legends.

3. Structural Analysis Limitations: Heavy reliance on AlphaFold3 predictions without experimental structure validation. The work will be enhanced to include at least one crystal structure or solution NMR validation of a redesigned intrabody.

We used AlphaFold3 to generate structural models as a prerequisite for inverse folding only. This negates the need for time consuming, expensive and sometimes unreliable experimental structural characterisation. Our aim with this work is to create an approach to intrabody generation that can be used by anyone with internet access and ~\$100 for gene synthesis. This aim is served by using publicly available tools with little, if any, overhead. This approach has allowed us to generate seventeen different intrabodies with validated interactions with some of the proteins that have been most challenging to medicine development, as noted in response to point 1 above.

No discussion of potential structural perturbations from extensive mutagenesis.

We have included a description of the substitution mutations incorporated by ProteinMPNN using the fixed position and model regime we implemented (section '*Impact of inverse folding with ProteinMPNN on scFv intrabodies*'). We found 90 % of substitutions are to highly or entirely surface exposed sites likely to have little effect on the molecular structure. Two in the VL domains are not solvent exposed but distant from the molecular core and two in the VH domain appear to increase core hydrophobic packing. The impact of the latter on VH domain thermal stability are noted.

The 0.02 Å vs 0.20 Å noise parameter choice lacks clear rationale.

The choice of 0.2 Å over 0.02 Å model noise resulted from empirical testing in anti-polyQ 3B5H10 and anti-SOD1 MS785 scFv intrabodies (**Fig. 4a**) and reflects published findings for ProteinMPNN, ProteinMPNN_{SOL} and LigandMPNN. An explanation of the relationship between model noise and sequence recovery and relevant citations has been added to the section '*scFv intrabody construction using inverse folding with solubility prediction*'. Furthermore, we made a detailed comparison of inverse folding with 0.02 Å and 0.2 Å model noise in section '*Impact of inverse folding with ProteinMPNNsol on scFv intrabodies*' including their impact on scFv charge.

4. Immunogenicity: ProteinMPNN designed FW sequences deviate from natural human antibody V genes, discussion around immunogenicity mitigation should be included.

This point has been included at the end of the *Discussion* section, including reference to a new ProteinMPNN variant reported to minimise immunogenicity.

5. Charge Calculation Oversimplification The (K+R)-(D+E) calculation ignores histidine contributions and local pKa shifts. Consider using more sophisticated electrostatic calculations.

Supplementary Table S1 shows R^2 values for intrabody solubility against charge at pH 5.5 for fifteen different levels of sequence from full-length scFv with tags down to single CDR loops. The Methods section '*in silico protein physicochemical analysis*' states how this was calculated and includes an approximation for histidine protonation in the formula $(K+R+H)-(D+E)$. We included a statement in the Discussion section noting that more refined surface electrostatic quantification may improve solubility prediction but, as noted above, our purpose is to democratise production and use of intrabodies by keeping the sequence design stage very simple.

6. Co-IP Controls Figure 6 co-immunoprecipitation experiments lack proper negative controls (empty vector, irrelevant scFv).

We have included a control MS785 immunoprecipitation with Z-FUS derived scFv as figure 6c.

- Supplementary files organization could be improved with a summary table.

Supplementary figures, tables and files are now included in an index on the first page of supplementary data.

- Some references are incomplete (e.g., ref 47, 48 listed as preprints)

These articles are currently only available as bioRxiv preprints.

- The scFvright website functionality should be described in more detail

scFvright makes a prediction of intrabody solubility from its net charge using the formula $\text{percent solubility} = -4.6237 \times \text{net charge} + 8.2469$. This is described in the Methods section '*scFvright website*'. PHP code has been made available as requested.

Reviewer #2

1. Benchmarking of the scFvright model.

The scFvright tool was trained on 45 scFv samples and yielded an R^2 of 0.75 between solubility and net charge. To support the robustness of this relationship, the authors should benchmark their linear model against existing solubility predictors such as Protein-Sol [1], DeepSol [2], or Protsolm [3]. Comparative analysis would confirm whether the proposed linear regression is sufficiently accurate and whether charge alone can capture most solubility variation.

A comparison of nine solubility predictors, including by Protein-Sol, is presented in **Figure S2**. We have made this clear in the text. We could not access the DeepSol server and were unable to install a functional version due to outdated dependencies. ProtSolM has been at preprint stage for 18 months. We tried to install but an unresolvable dependency in the Conda yaml file and deprecated filter_backbone function means the GitHub version is currently unusable.

2. Feature interaction analysis.

Although the manuscript reports that hydrophobicity, pLDDT, and other physicochemical features individually correlate weakly with solubility, it would be valuable to assess possible interaction effects with charge. A multivariate regression or feature interaction model could reveal whether, for instance, charge-hydrophobicity coupling provides additional explanatory power.

The reviewer is correct there is almost certainly an impact of hydrophobicity and other

molecular characteristics on solubility as has been demonstrated many times for other proteins. However, we note the very similar quality of predictions generated by Prot-Sol (**Fig. S2**) and scFvright. Prot-Sol, using multi-parameter prediction, would very likely perform better on a more heterogenous group of proteins but for structurally highly conserved scFv there is little difference. Redesigning for net charge and allowing ProteinMPNN to improve thermal stability is sufficient to allow good expression, availability and target interactions; what we set out to do.

3. Clarification of scFvright's purpose

The scFvright web tool appears to serve only as a validation of the empirical charge–solubility relationship. Since it is not subsequently used in the AI design workflow, its independent contribution to the study is unclear. The authors should clarify whether scFvright is intended solely for exploratory prediction or as part of the broader intrabody engineering pipeline.

scFvright is a standalone web tool to predict scFv solubility from its primary sequence. It has no role in protein redesign. It could be used to predict the solubility of ProteinMPNN output sequences but that operation could also be performed in Excel, Matlab etc. We have not amended the text on this point because scFvright is introduced in a section unrelated to protein redesign.

4. Interpretability of ProteinMPNNSOL results

The inverse folding section demonstrates improved solubility and thermal stability, but lacks analysis explaining why certain AI-generated sequences perform better than others. Comparing top-ranked (e.g., top 1%) versus low-ranked (e.g., bottom 50%) sequences could identify key sequence or structural determinants—such as altered surface charge, salt bridges, or hydrophilic patches—that underlie the observed performance gap.

We thank the referee for this perceptive statement; it may be there are more relationships that tie primary sequence to scFv intrabody charge and solubility. However, our initial experiments indicate these relationships are not easily recognised and will require a large amount of construct synthesis and testing along with new computational tools to analyse the data. In this paper we have described the biggest factor that determines solubility and this can be applied as a profiling tool to both native and inverse folded sequences.

5. Reproducibility and workflow transparency

The AI redesign approach is innovative but under-documented. A complete workflow should be provided, detailing the parameters used (e.g., fixed residues, linker constraints, noise level, selection criteria, number of generated variants). Supplementary tables or pseudo-code would substantially improve reproducibility and future applicability of ProteinMPNNSOL for antibody optimization.

Section '*High throughput in silico scFv intrabody construction*' specifically deals with this subject and includes fixed residues in standardised Chothia amino acid numbering, linker constraints and noise level. We have also amended this section to include our selection criteria, a call-out for the graphic representation of our method in **Figure 5a** which is a natural language pseudo-code for antibody optimisation with ProteinMPNN, and the number of sequences output by ProteinMPNN.

6. Integration of functional prediction tools

While fixing framework or linker regions helps preserve antigen binding, it also limits AI design freedom. The authors could enhance the design pipeline by integrating antibody binding or affinity prediction tools to prescreen ProteinMPNNSOL generated variants for binding competence. This would establish a more complete and automated workflow from sequence design to functionality prediction.

This is a great suggestion, and is the focus of our current grant applications, but is a project in itself. The advantage of the approach documented in our paper is that epitope-paratope interactions are maintained from parent antibody to redesigned intrabody by conserving both CDR loops and variable domain interfaces. We are not redesigning the whole molecule, which would justify an *in silico* affinity maturation step, just the non-paratope surface and selecting those candidates that have high predicted solubility.

Kindest regards,

Dr Gareth Wright
School of Life Sciences
University of Essex
Wivenhoe Park
Colchester
Essex
CO4 3SQ
United Kingdom

Email: gareth.wright@essex.ac.uk